# Impacts of climate on the biodiversity-productivity relationship in natural forests

Songlin Fei [1], Insu Jo [1], Qinfeng Guo [2], David A. Wardle[3,4], Jingyun Fang[5], Anping Chen [1], Christopher M. Oswalt[6] & Eckehard G. Brockerhoff [7,8]

Understanding biodiversity-productivity relationships (BPRs) is of theoretical importance, and has important management implications. Most work on BPRs has focused on simple and/or experimentally assembled communities, and it is unclear how these observed BPRs can be extended to complex natural forest ecosystems. Using data from over 115,000 forest plots across the contiguous United States, we show that the bivariate BPRs are positive in dry climates and hump-shaped in mesic climates. When considering other site characteristics, BPRs change to neutral in dry climates and remain hump-shaped in humid sites. Our results indicate that climatic variation is an underlying determinant of contrasting BPRs observed across a large spatial extent, while both biotic factors (e.g., stand age and density) and abiotic factors (e.g., soil properties) can impact BPRs within a given climate unit. These findings suggest that tradeoffs need be made when considering whether to maximize productivity vs. conserve biodiversity, especially in mesic climates.

[1] Department of Forestry and Natural Resources, Purdue University, West Lafayette, IN 47907, USA. [2] Eastern Forest Threat Assessment Center, Forest Service, US Department of Agriculture, Research Triangle Park, NC 27709, USA. [3] Asian School of the Environment, Nanyang Technological University, Singapore 639798, Singapore. [4] Department of Forest Ecology and Management, Swedish University of Agricultural Sciences, 90187 Umea, Sweden. [5] Institute of Ecology and College of Urban and Environmental Sciences, Peking University, 100871 Beijing, China. [6] Southern Research Station, Forest Service, US Department of Agriculture, Knoxville, TN 37919, USA. [7] Scion (New Zealand Forest Research Institute), Christchurch 8440, New Zealand. [8]Present address: Swiss Federal Research Institute WSL, Zürcherstrasse 111 CH-8903 Birmensdorf, Switzerland. These authors contributed equally: Songlin Fei, Insu Jo. Correspondence and requests for materials should be addressed to S.F. (email: sfei@purdue.edu)

The biodiversity-productivity relationship (BPR) has been one of the central topics in ecology due to its theoretical importance and management implications. A wide variety of BPRs have been reported from field studies and meta-analyses, including linear positive and negative, concave positive and negative, and neutral forms[1–7]. In general, results from small-scale experiments that have directly manipulated species richness, mostly of herbaceous plants in grasslands, have shown that BPRs are positive, but decelerating as species richness increases[2,6]. On the other hand, observational studies that have looked at the correlations between richness and productivity in natural systems have found various forms of BPRs[5,7]. This inconsistency in reported BPRs has often been ascribed to differences among studies in factors including spatial grain and extent, sampling methods, ecosystem types, and taxonomic groups[8–11]. Importantly, the majority of BPR research has been based on herbaceous communities[4–7]. Relative to the tremendous number of publications on grassland BPRs, investigations on BPRs in forest communities are comparatively few. A few studies have attempted to understand forest BPRs with broad-scale observational datasets, but have reached divergent conclusions. While positive BPR has been found in some studies[12–14], others BPR types have also been reported such as hump-shaped, neutral, or negative[13,15,16].

Over broad spatial scales, regional climate is widely regarded as a determinant of both biodiversity and productivity patterns. The role of climate in influencing the shape of BPRs, however, has received little attention. In general, productivity has a positive linear relationship with temperature in cold regions[16] and with precipitation in dry ecosystems[17,18]. Biodiversity usually increases with temperature and precipitation[19,20]. However, there is also evidence that in some instances, increasing temperature and water availability favors more competitive species, which may reduce species coexistence and therefore diversity[21,22]. Because productivity and biodiversity may show partially differing relationships with climate, different BPRs could conceivably occur in different regions across a large geographical area (e.g., continental or subcontinental). Furthermore, other abiotic factors (e.g., soil fertility and disturbances) and biotic factors (e.g., forest characteristics) are likely to have impacts on biodiversity and productivity, and thereby modify BPRs. Grace et al.[4] devised a structural equation model (SEM) that explicitly separates the effects of climate, soil, and disturbances on both productivity and species richness across 39 grassland sites across the world. They found that richness increased with precipitation in the warmest season, and that climate played an important role in controlling variation in productivity. However, they pooled all plots in a single SEM model, so it is unclear how BPRs may have changed across different climatic ranges. To date, no study has attempted to explicitly address how BPRs may change across climatic gradients or across contrasting climates at a large spatial extent.

In the present study, we examine BPRs in forests across a large range of temperature and precipitation conditions in the United States (US), using data from the US Forest Inventory and Analysis (FIA) program [https://www.fia.fs.fed.us/]. The FIA data, including species composition, diameter, height, age, and many other attributes, were systematically collected across the contiguous 48 states of the US (excluding Alaska and Hawaii). Because of the broad geographic coverage and systematic design, FIA data have been widely used in various regional studies[23–25]. Our analysis included a total of 115,578 plots (each 672 m² in area) at a spatial sampling intensity of one plot per 2428 ha across the entire contiguous US (Supplementary Fig. 1), sampled during the 2012–2016 inventory cycle. These plots spanned across 194 ecoregions (see Supplementary Fig. 1), and all trees with diameter at breast height larger than 5.0 cm have been recorded in each

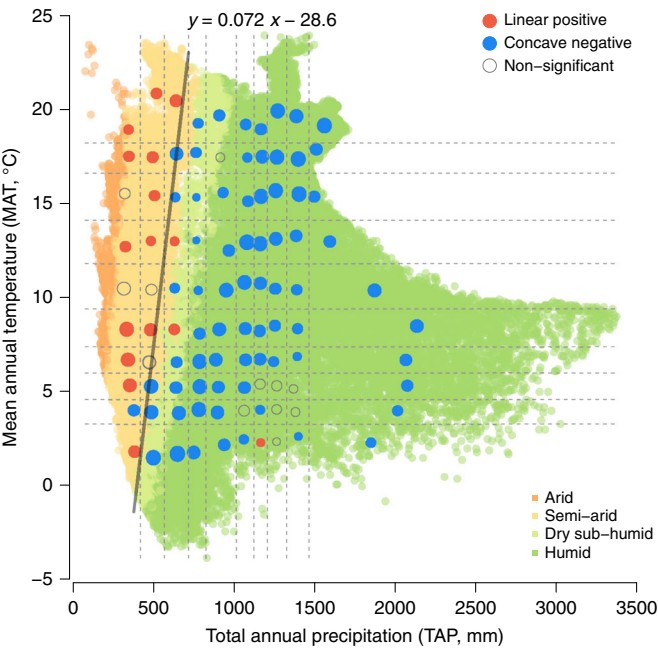

**Fig. 1** Relationship between tree richness and productivity in different climatic units. The color of circles within each climatic unit indicates the form of the relationship determined by a generalized linear model between biodiversity and productivity, and circle radius is proportional to the number of forest plots (log-transformed) in each climatic unit. Forest plots were divided into 10 × 10 climatic units according to their MAT and TAP quantile classes based on WorldClim[25]; and are colored according to their aridity index (0.03–0.2, arid; 0.2–0.5, semi-arid; 0.5–0.65, dry sub-humid; > 0.65, humid) based on the Global Aridity Index[26]. The line between the red and blue points is the division between the linear-positive and concave-negative relationships based on a logistic regression for the two groups as a function of MAT and TAP

plot. Similar to Liang et al.[13], we used mean annual increment in tree biomass (total above-ground live biomass divided by stand age) as a measure of forest productivity and tree species richness as a measure of biodiversity to test BPRs.

We show that both linear-positive and concave-negative bivariate BPRs exist in natural forests depending on the underlying climatic. With the consideration other biotic and abiotic factors, BPRs become non-significant in harsh climates and concave-negative in humid climates. Our study indicates that climatic variations can be an underlying determinant for the contrasting BPRs reported previously among different studies.

## Results

**BPR patterns in different climates**. To identify how apparent BPR patterns may vary across climatic space, we defined 10 quantile classes for both mean annual temperature (MAT; range −3.8–23.9 °C) and total annual precipitation (TAP; range 79–3375 mm) based on the distribution of these forest plots in the MAT vs. TAP climatic space, resulting in 100 climatic units (10 × 10 grids in Fig. 1; see Supplementary Data 1 for detailed quantile classes).

By analyzing plots located within each of these climatic units with generalized linear models (glms), we found that the bivariate BPR within a climatic unit was determined primarily by the climatic space in which these plots reside (Fig. 1 and Supplementary Fig. 2). The relationship between tree species richness and log-transformed productivity was linear-positive in arid and semi-arid climatic units, and concave-negative in humid and sub-humid units (Fig. 1). No other significant BPR types

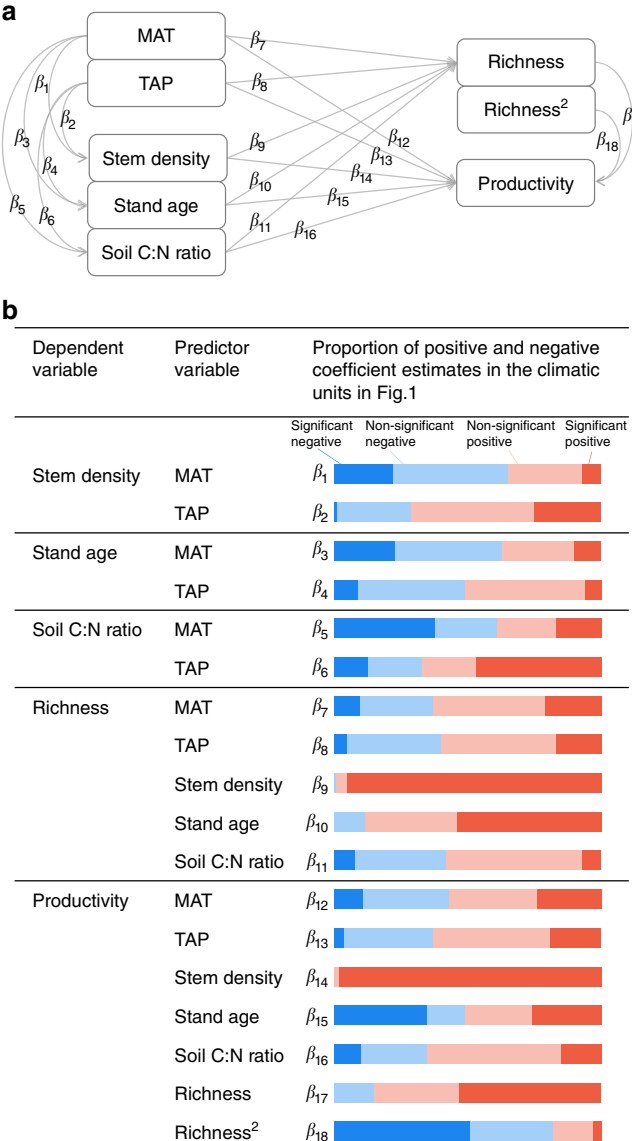

**Fig. 2** A hierarchical model for testing effects of environmental factors and richness on productivity. **a** Model structure and **b** proportions of positive and negative coefficient estimates ($\beta$s) across the climatic space. Significance of the coefficient is evaluated based on whether the 95% credible interval (CI) overlaps zero. A list of coefficient estimates with its 95% CI is available in Supplementary Data 2. MAT mean annual temperature, TAP total annual precipitation

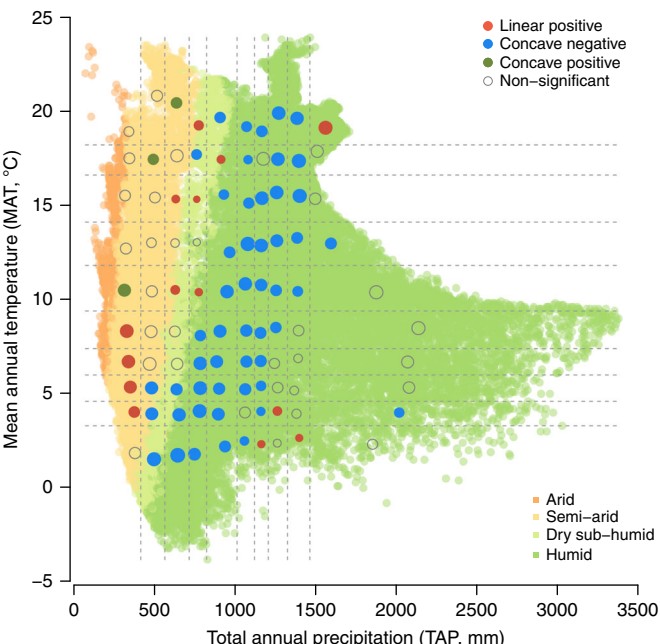

**Fig. 3** Relationships between tree richness and productivity in different climatic units, determined by coefficients ($\beta_{17, 18}$) estimated from a hierarchical Bayesian model illustrated in Fig. 2. The circle radius is proportional to the number of forest plots (log-transformed) in each climatic unit. A list of the coefficient estimates with its 95% CI is available in Supplementary Data 2

further test the BPRs across plots within each of these climatic units. The model contains five sub-models with eight variables and 18 coefficient relationships (Fig. 2a; also see Methods)). To deal with potentially non-linear relationships, as indicated in Fig. 1, we also included a quadratic term of richness in the model (Fig. 2a). Since plots of distinct geographical regions in this study may be pooled together in the climatic unit because of their shared climatic space, we added 'ecoregion' (a geographical entity in which all elements share a similar biological and environmental history[26]) as a random intercept in the model. The results indicated that both biotic and abiotic factors impacted on BPRs across the climatic space (Fig. 2b), as recognized previously[4,13]. Although the BPR forms were changed relative to the bivariate analyses for certain climate regions (Fig. 1 vs. Fig. 3), the separation of different BPR forms remained apparent in the climatic space. Compared to the results from bivariate analyses (Fig. 1), BPRs in arid and semi-arid regions (upper-left) changed from mostly positive to mostly neutral (i.e., non-significant); BPRs in cold-mesic regions (lower-right) changed from hump-shaped to mostly neutral; and BPRs in the humid regions (middle section) remained hump-shaped ($\beta_{17-18}$ in Figs. 2b, 3).

Climate directly impacted the observed BPR patterns, which were expressed through differing associations of climatic variables vs. richness and climatic variables vs. productivity in different regions of the climate space (Fig. 4). In general, MAT had a negative but non-significant association with richness within each climatic unit for the arid and semi-arid regions, a significant positive association for cold-mesic regions, and a significant negative association for hot-mesic regions (Fig. 4a). The associations of MAT vs. productivity were mixed in the arid and semi-arid regions, but had a similar pattern to MAT vs. richness in the mesic region (Fig. 4c). The associations of TAP vs. richness and TAP vs. productivity were similar within each climatic unit across the climate space (Fig. 4b, d). In general, richness and productivity had positive associations with TAP in

(e.g. linear negative, concave positive) were found. Only 12 of the 100 relationships were not statistically significant, which is likely due to relatively low sample sizes compared to other climatic units ($n < 600$ plots for 10 of these 12 units). A logistic regression of the two BPR types (linear-positive vs. concave-negative) against MAT and TAP showed a clear separation between the two BPR types in the climatic space, for which the logistic regression line falls near the border of the semi-arid and dry sub-humid climate zones (Fig. 1). In general, linear-positive BPRs were located to the left and concave-negative BPRs to the right of the regression line.

**Effects of biotic and abiotic factors on BPRs.** To understand how other biotic and abiotic factors can affect BPRs across the climatic space, we used a hierarchical Bayesian framework to

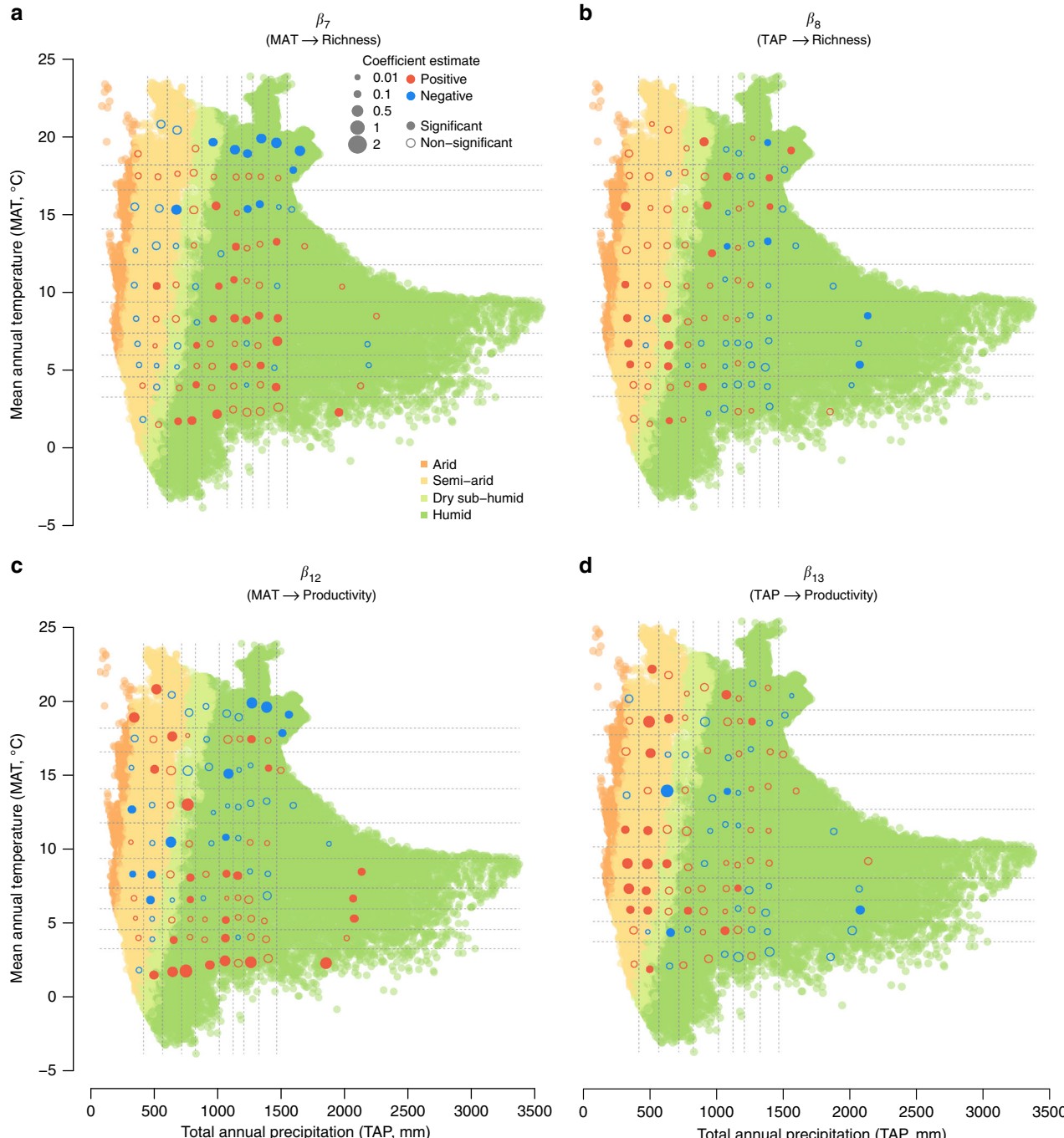

**Fig. 4** Impacts of climate on richness and productivity in different climates. **a** Impacts of MAT on richness, **b** impacts of TAP on richness, **c** impacts of MAT on productivity, and **d** impacts of TAP on productivity. Coefficient estimates of the relationship for each climatic unit are plotted based on the means of MAT and TAP within the climatic unit on the MAT-TAP space illustrated in Fig. 1. Significance of the coefficient is evaluated based on whether the 95% credible interval (CI) overlaps zero. A list of coefficient estimates with its 95% CI is available in Supplementary Data 2

dry climatic units, but negative associations in mesic climatic units. Overall, there were more climatic units that had statistically significant associations (at 95% credible interval) between MAT and richness (31 units) and between MAT and productivity (35 units) than between TAP and richness (22 units) and between TAP and productivity (23 units).

Climate also had indirect impacts on BPR patterns via their varying associations with certain biotic and abiotic factors in different regions of the climatic space ($\beta_{1-6}$ in Fig. 2b), and via the different associations of these factors with richness and productivity in different regions of the climate space ($\beta_{7-16}$ in

Figs. 2b, 5). Across the climatic space, and particularly in dry climatic units, stem density was negatively associated with MAT and positively with TAP ($\beta_{1,2}$; Supplementary Data 2), while stand age and soil C:N ratio had no clear patterns in the climatic space with the two climate variables ($\beta_{3-6}$; Supplementary Data 2). Some of these site characteristics in turn had significant and sometimes different associations with productivity and richness in different regions of the climatic space. In general, stem density had positive associations with both richness and productivity (Fig. 5a, c). The impacts of stand age on richness and productivity were variable (Fig. 5b, d). Overall, stand age

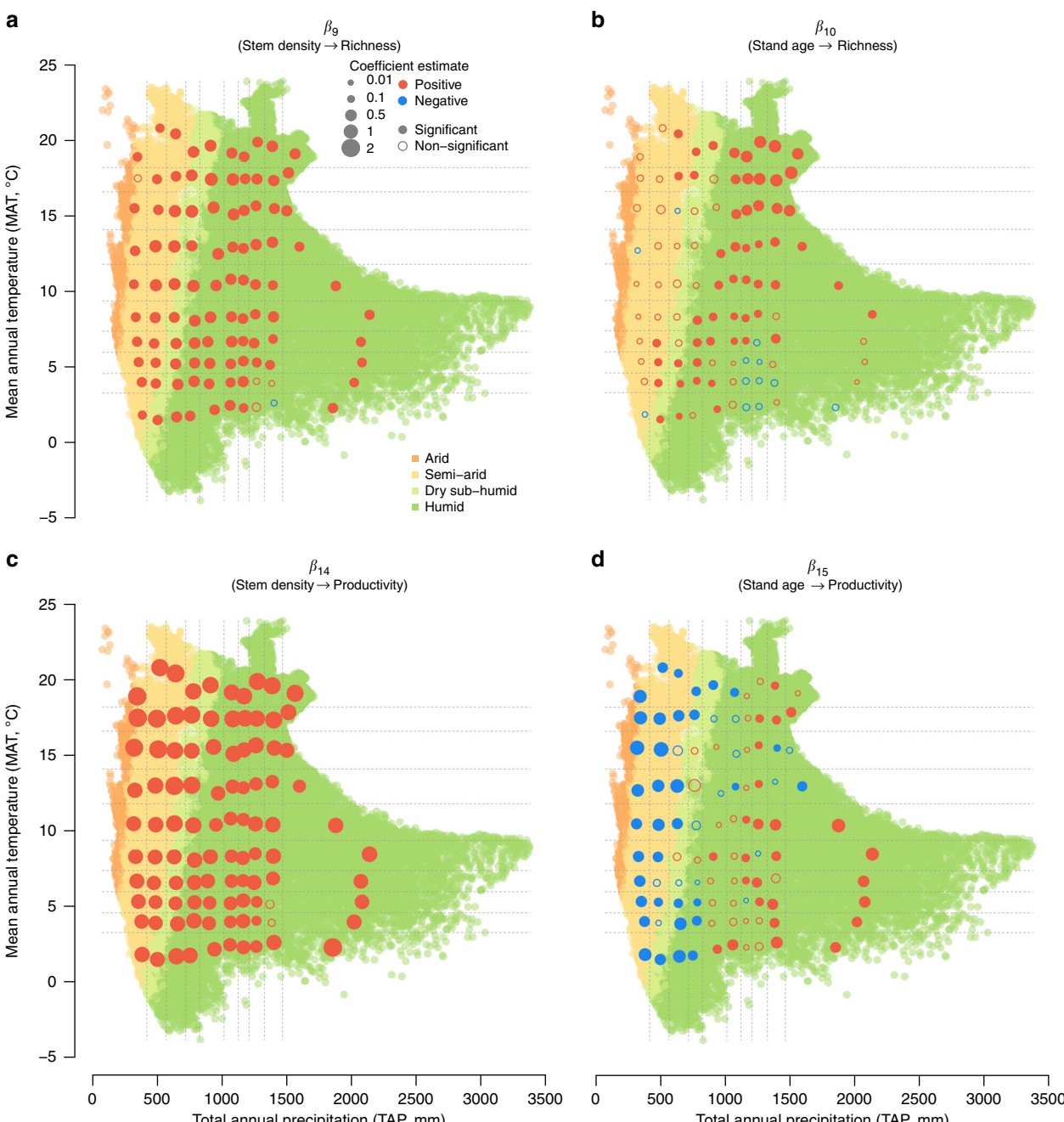

**Fig. 5** Impacts of stand characteristics on richness and productivity in different climates. **a** Impacts of stem density on richness, **b** impacts of stand age on richness, **c** impacts of stem density on productivity, and **d** impacts of stand age on productivity. Coefficient estimates of the relationship for each climatic unit is plotted based on the means of MAT and TAP within the climatic unit on the MAT-TAP space illustrated in Fig. 1. Significance of the coefficient is evaluated based on whether the 95% credible interval (CI) overlaps zero. A list of coefficient estimates with its 95% CI is available in Supplementary Data 2

had positive associations with richness within each climatic unit, but the associations were not statistically significant in the arid, semi-arid, or cold-mesic regions (Fig. 5b). The associations between stand age and productivity were negative in arid and semi-arid regions and positive in mesic regions (Fig. 5d).

## Discussion
Our study provided a climatic perspective for understanding variation in the shapes of BPRs observed in natural forests. Conclusions from previous manipulative experiments involving synthetically assembled communities may not be easily extrapolated to the real world where climate-dominant regional variations may impact on observed BPR patterns. Our results suggest that both linear-positive and concave-negative bivariate BPRs can be widely observed in natural forests, but that they dominate in different climatic space. When other biotic and abiotic factors were taken into consideration, BPRs were primarily non-significant in harsh climates (such as dry or extreme wet or cold climates) and the predominant BPR form was concave-negative in humid climate. Therefore, all previously reported BPRs are plausible, but only within the given climate where

measurements were taken and when other biotic and abiotic factors are taken into consideration. We show that climatic variations can be an underlying determinant, both directly and indirectly, of the contrasting relationships between productivity and biodiversity that have been observed between different studies. Climatic variation helps with the partitioning of linear vs. hump-shaped BPRs across climatic space, and assists in explaining the various relationships between productivity and other site characteristics observed within each climatic unit. The shifts in BPRs with or without the consideration of site characteristics illustrate the importance of building comprehensive models when studying BPRs, as has been suggested by earlier studies[4,13]. However, further field validation will be required to determine if our findings may be translated to other ecosystem types. In addition, as BPRs can be influenced by spatial scales of observation[11], results from this study are relevant at the large spatial scale we considered here, and it is likely that other relationships emerge at other scales.

The observed changes in BPRs from arid to mesic regions have plausible theoretical explanations. Some plant strategy theories[27,28] suggest that different BPRs could be derived based on the shift of the dominance from facilitation (which drives the formation of a positive BPR) in low-productivity sites to competitive exclusion (which favors a few highly productive and competitive species dominating the community) in high-productivity sites. Alternative theories indicate that BPRs may be influenced by 'complimentary resource use' among coexisting species in a community, and by 'selection effect' due to individual species or groups of species that have disproportionate effects on community productivity[6,29]. When other biotic and abiotic factors were taken into account, there were no significant relationships between richness and productivity in relatively harsh environments, which points to no obvious role of facilitation in those environments. In contrast, the hump-shaped BPRs in relatively productive sites is consistent with the occurrence of both competition and facilitation for different portions of the relationship[30,31]. However, we do not currently have an effective way of differentiating between the role of sampling effect and complementarity because sites in comparative studies across different forest types and climate ranges do not share a common species pool.

Our findings have important implications for management, conservation, and restoration of forests. Forests harbor a rich biodiversity and provide many essential ecosystem services, but they are threatened globally by deforestation, climate change, species invasions, and other factors. In addition to climate factors, we have shown that forest characteristics can impact the BPR patterns observed. In the US, most of the forests have been modified by human impacts to varying extents, and the majority of US forests are still in recovery from long-term disturbances, making the average forest age relatively young[32]. Our results indicate that as forests age, richness keeps increasing in almost all climates, especially for mesic sites. As biodiversity is closely linked to many other ecosystem functions[33,34], this enhanced forest tree species richness with increasing stand age highlights the importance of old-growth forests in biodiversity conservation. Further, the recovery of US forests over the past decades has made them a significant atmospheric carbon sink[35,36], although the potential of this carbon sink under future climate and emission scenarios is uncertain. Meanwhile, as forests also play a critical role for both biodiversity conservation and carbon sequestration, weighting the relative importance of the two is an important consideration for forest management, and an improved understanding of forest BPRs will help inform forest management decisions. In addition, as temperature rises and precipitation shifts continue through global climate change, shifts in BPR forms are likely to occur, especially in the transition zones between semi-arid

and sub-humid regions. Finally, our findings suggests that to manage and/or restore forest biodiversity, different strategies are needed in different climate regimes. At least in the US, maximum tree diversity and productivity can be achieved simultaneously for regions where the climate is mesic, but tradeoffs need to be made in maximizing biodiversity vs. productivity when climate is arid.

## Methods

**Forest inventory data.** Inventory data were collected from the US Forest Inventory and Analysis (FIA) program (US Forest Service, data available at [https://www.fia.fs.fed.us/]). The FIA program monitors spatiotemporal patterns of forests resources at the national level, using a fixed grid of permanent plots, which have a sampling intensity of approximately one plot every 2428 ha. Each plot is 0.067 ha, and comprises four smaller fixed-radius (7.32 m) subplots spaced 36.6 m apart in a triangular arrangement with one subplot in the center. Tree-level attributes such as diameter at breast height (dbh) and species are measured for all stems with dbh > 5.0 cm, and site-level attributes such as stand age are also measured. For each plot, we extracted total above ground biomass, stand age, and richness (total number of tree species), stem density information from the FIA database, with most data collected through the 2012–2016 inventory cycle. We used mean annual increment in tree biomass (total above ground biomass divided by stand age) to estimate forest productivity[13], and tree species richness to represent biodiversity. Due to the limitation of data availability, our productivity calculation does not include below-ground productivity (roots or associated mycorrhizae), which may also be influenced by species richness and climate.

**Climate and soil data.** At the plot-level, mean annual temperature (MAT) and total annual precipitation (TAP) were derived from Global Climate Data - WorldClim Version 1.4 (30 arc-second resolution; available at [www.worldclim.org][37], aridity index (a ratio of mean annual precipitation over mean annual potential evapotranspiration) was derived based on the Global Aridity Index (30 arc-second resolution; available at [www.cgiar-csi.org/data][38], and soil C:N ratio in 0 to 20 cm depth (as a measure of soil fertility) was derived based on World Soil Information (30 arc-second resolution; available at [http://www.isric.org][39]. We excluded plots with missing values, and ended up with 115,578 plots for analyses.

**Statistical analysis.** To assess the biodiversity-productivity relationship (BPR) across the climatic space, we defined 10 quantile classes for both MAT and TAP based on the distribution of plots in the climatic space, resulting in 100 climatic units (10 × 10 grids in Fig. 1). Using plot-level data (tree species richness and productivity) within each climatic unit, we modeled productivity (log transformed) as a function of richness using a Gaussian generalized linear model (glm). We compared a model with both linear and quadratic terms of richness (quadratic model) and a model with only a linear term of richness (linear model), then selected the best model that has a lower AIC to determine BPR types (linear or concave). If the linear term was not significant ($P > 0.05$; $Z$-test) in the linear model or the quadratic term was not significant in the quadratic model, we reported it as a non-significant relationship. Summary statistics and bivariate relationship figures with the fitted line for each climatic unit are available in Supplementary Data 1 and Supplementary Fig. 2. The spatial boundary between the BPR types (linear positive and concave negative) in the climatic space was determined based on a logistic regression of the BPR types as a function of MAT and TAP (Fig. 1).

Using a hierarchical Bayesian modeling approach, we tested the robustness of the bivariate BPR patterns observed in climatic space and the impacts of other key biotic and abiotic factors on productivity. The model included sub-models with five dependent variables: stem density, stand age, soil C:N ratio and productivity (all with normal distributions) and richness (with a Poisson distribution) and estimated 18 posterior coefficients ($\beta_{1-18}$) for the relationships among these variables (Fig. 2a, Supplementary Data 2). Based on estimated posterior coefficients of linear and quadratic terms of richness ($\beta_{17, 18}$) in the productivity sub-model, we determined the BPR type for each climatic unit (Fig. 4): linear positive if $\beta_{17} > 0$, $P(\beta_{17} > 0) > 0.95$, and 95% credible interval (CI) for $\beta_{18}$ overlaps with zero; concave negative if $\beta_{18} < 0$ and $P(\beta_{18} < 0) > 0.95$; concave positive if $\beta_{18} > 0$ and $P(\beta_{18} > 0) > 0.95$; and non-significant if 95% CIs for both $\beta_{17}$ and $\beta_{18}$ overlap with zero. We used stem density and stand age to express stand characteristics, MAT and TAP to represent within unit climate variability, and soil C:N ratio as a measure of soil fertility (in line with[13]). As these variables can be influenced by other regional processes and site history, we added ecoregions in the sub-models as a random intercept to account for the spatial heterogeneity between geographically distant plots that share a similar climate. Ecoregions were delineated based on historical factors (i.e., past climate and landform) and forest composition and structure (which were set to be relatively uniform within each ecoregion)[26]. Stem density, stand age, soil C:N ratio, and productivity were log-transformed to meet normality assumptions, and all predictor variables were standardized by subtracting its mean and dividing by two standard deviations to make effect sizes comparable among the variables[40]. We used non-informative priors for intercepts and slope coefficients ($\beta$s) of the relationships in Fig. 2a from a normal distribution of mean = 0 and variance = 1000. Model structure and components are listed in Supplementary

Data 2. The model was fitted using Markov chain Monte Carlo methods (MCMC) in JAGS in R 3.3.1 with three parallel MCMC chains for 100,000 iterations with a 5000-iteration burn-in[41–43].

**Reporting summary**. Further information on experimental design is available in the Nature Research Reporting Summary linked to this article.

## Data availability

Data used in this study are available at an open data repository (Purdue University Research Repository, https://doi.org/10.4231/R7SB440D)[44].

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

## Acknowledgements

Thanks to hundreds of FIA field crew members for the data used in our study. The study was partially supported by NSF (#1638702) and USDA McIntire-Stennis program to S.F.

## Author contributions

S.F. conceived the study and drafted the manuscript. I.J. and C.M.O. compiled the data. I.J. performed the analyses. I.J., Q.G., D.A.W., J.Y., A.C., C.M.O. and E.G.B. contributed substantially to the interpretation of the results and the writing of the manuscript.

## Additional information

**Competing interests:** The authors declare no competing interests.

