## [Peer Review File · Nature Communications]

Reviewers' comments:

Reviewer #1 (Remarks to the Author):

Overall the Authors present a very interesting and important study on the biodiversity-productivity relationship in natural forest and the importance of climate as predictor of this relationship. I have a couple of major points that I think could assist with establishing the important context of the study and several other issues that I hope will increase the clarity and depth of ecological meaning that can be drawn by readers.

I am not sure I agree with how the Authors begin framing the novelty of the study. I think there has been a tremendous number of studies investigating tenets of the BPR and a lot of evidence has mounted but the evidence is equivocal especially for more complex ecosystems like forests--- simpler communities such as experimentally derived mesocosms of algae, herbaceous plants etc. have provided I think many generalizable constructs but these have not necessarily held true when then applied and tested in more natural systems etc. I think this is a novel contribution of the Authors study and they should focus on this, and their incorporation of climatic predictors---- bringing a reality check so to speak to this important relationship.

I think the Authors need to re-write their abstract so that it explicitly delineates what the observed divergent relationships have been in natural systems and specifically in forests versus grasslands (where a preponderance of the studies were to date conducted) and then describe explicit what the new findings that this study contributes especially in regards to climatic variation and how precipitation potentially regulates this relationship.

Line 45 to 47: It would be good to describe briefly the key competing theories...if the Authors study will expand on these ideas we need to see a "thesis" of what they think/interpret these are and I think there should be a focus on natural communities.

Line 54: what are these exceptions? With increasing resources, there is evidence of a selection for more competitive plant species for these resources.

Line 57-58: In natural forest ecosystems but it has been done in grasslands by Grace et al. 2016 and you cite this study earlier...what did they find? Illustrate your points to readers with brief and meaningful examples...your readers should not have to go and read other study to find your examples or what the debates etc are about...the Authors should be illustrating these ideas for readers in a complete study. This is an important study for your paper and the supplementary material includes a very detailed and insightful history of the literature around this topic.

Line 59: why the reference to the USA being contiguous...shouldn't the emphasis be on the range of climates etc to fit the purpose of the study.

Line 61: a comma before which

Line 59-67: Great dataset...are these all natural forests or were some of these plantations. Logged and then regrown especially given the data is from the US Forest Inventory. When was the data collected---all in the same year or over many years? Are the study sites all different forests? How many species do they hold in common...why was the inventory taken on all of these forests from an ecological stand point what ties them together—is there a management history in common? I can see now that some of this information is provided in the supplementary material but I think some of it is key for understanding the ecological meaning and robustness of the Author's study and therefore should be presented briefly in the main manuscript.

Line 63-67 what about basal area as a measure instead of biomass...can you provide a justification for why you chose this particularly measure of productivity?

Line 73-74: given the complexity of the data and the sampling design, wouldn't it be more robust to use generalized mixed models. This would allow the Authors to repartition the variation in the data according to sites and plots; thereby incorporating the sampling design into the analytical models. I can see that you do use GLMMs later on but why not throughout?

Line 100-101 Could this relationship found between MAT and productivity when TAP is >1000mm be because there is a little variability in MAT in your dataset at such high precipitation levels. What is the relationship between MAT and TAP?

Have you thought about using a structural equation model approach like what Grace et al. 2016 Nature used for grasslands?

Line 139: suggest rewording with "...depending on the relative influence of species level interactions such as facilitation and competition."

Line 143: I don't think you have any way of differentiating between the sampling effect and complementarity because there may be no commonality between species identity amongst the different forest types or are there? Nevertheless, the Authors do not illustrate this for their readers.

Line 155-169: I think the management implications of this study come from nowhere with providing readers with an understanding of these forests...have they been logged and regrown etc.

Line 261 I think the Authors should include a definition for the black color in panel A of Fig. 2 with the legend description provided for all other colors.

Fig. 2: There are quite a number of different predictors in this model. Were predictors scaled? What were the R² values with and without random effects for these models, what were the denominator and numerator degrees of freedom?

Reviewer #2 (Remarks to the Author):

The main purpose of this study is to understand what drives the form of the Biodiversity-Ecosystem functioning (BEF) relationship which has been one of the main topics in ecology in the past decades. The main hypothesis is that the contrasting results observed in experimental settings and in observational studies may be explained by climatic factors. The results obtained by this study that analyzed natural forests ecosystems in the contiguous United States is that BEF relationship is determined by a combination of total annual precipitation and mean annual temperature, because the relationship between richness and log(productivity) is linear-positive in arid and semi-arid sites, and concave-negative in humid and sub-humid sites. The authors arrived to this conclusion by analyzing data from more than 116,000 plots across forests from which estimations of aerial primary productivity and species richness are available, so this is an observational study and the mechanism behind this pattern cannot be inferred so far. In addition, the ecosystem functioning variable has been log-transformed (log productivity is reported along the manuscript) whereas the richness hasn't, so the inferences relating linear or concave negative functions might be confusing for the broad audience. And the comparison with previous studies need careful at this point.

The consequences of biodiversity losses on plant production were previously evaluated, but regional studies combining diversity and climate effects are scarce. In addition, the BEF literature was developed based on grasslands experiments involving herbal species, but has rarely been assessed in forests. In synthesis, the study was well designed and the results are compelling and

can potentially make an important contribution to the effect of climate on productivity and richness in forests, but the improvement in the knowledge of the BEF relationship is much more limited, and the conclusions cannot be translated to other ecosystems such as grasslands or deserts. I made some suggestions on how to improve the manuscript in my comments bellow.

Specific comments

- The main subject of this study is the Biodiversity-Ecosystem functioning relationship, which has been mentioned as BEF in the literature (see for example Flynn et al. 2001 or Tilman et al. 2014), even when the main functioning aspect assessed was "primary productivity" and the diversity index was "species richness". So, I suggest keeping this terminology, changing biodiversity-productivity relationship (BPR) by Biodiversity-Ecosystem functioning (BEF) along the Ms.
- #26. I don't think the following phrase is totally true: "However, we still lack an understanding of what drives the form of the BPR or an underlying theory to explain the varying patterns of BPR across studies". We do have a good understanding and theory of the BEF (see for example Tilman et al 2014 for a synthesis). What we do lack is the influence of climate. And the application of the BEF relationship at larger (regional) scales. I prefer the statement "Nevertheless, to date no large-scale studies have attempted to explicitly address how BPR may change across climatic gradients".
- #31 "We found that the relationship between richness and log(productivity) is linear..." What does this statement mean in terms of the productivity-richness relationship? How do you translate the log-productivity patten to the untransformed productivity variable?
- #35 Productivity generally increases with precipitation, while richness has a concave negative relationship with precipitation. It depends on the scale!!! (Chase and Leibold 2002). And on the range of precipitation analyzed (as is acknowledge latter in the Ms). The productivity-precipitation relationships saturated after 1,500 mm/yr precipitation probably because other variables than water limits plant production.

References

- Chase, J. M. & Leibold, M. A. Spatial scale dictates the productivity-biodiversity relationship. 179 Nature 416, 427-430 (2002).
- Flynn DFB, Mirotnick N, Jain M, Palmer MI, Naeem S. 2011. Functional and phylogenetic diversity as predictors of biodiversity-ecosystem-function relationships. Ecology 92:1573-81
- Tilman et al 2014. Biodiversity and Ecosystem Functioning. Annu. Rev. Ecol. Evol. Syst. 45:471-93

Reviewer #3 (Remarks to the Author):

The goal of this study is to determine the cause of variation in diversity-productivity relationship. The authors begin with the claim that diversity-productivity relationships are highly variable across space, sometimes being linear, other times being curvilinear (concave down). They proceed to characterize the diversity-productivity relationships for trees in U.S. forests using estimates of species richness and productivity of trees in 0.01 sq. km plots inventoried for the U.S. Forest Inventory and Analysis (FIA). They then attempt to explain variation in the diversity-productivity relationship based on environmental variables, namely air temperature and precipitation, that are taken from ~1-km gridcells represented in the Global Climate Data database. Lastly, the authors examine how the diversity-productivity relationships vary by across ecoregions of the U.S. (I assume these were EPA Level III ecoregions), presumably because these represent distinct climatic environments.

In my view, the study is presently limited by three problems:

1. Setting up a straw-man argument. To set up their study and justify its importance and contribution to ecology, the authors make two claims: (1) "A wide variety of BPRs have been reported from field studies and meta-analyses" (line 42) and (2) "we still lack an understanding of what drives the form of the BPR or an underlying theory to explain the varying patterns of BPR

across studies" (line 26).

As evidence for these two claims, the authors cite papers from two distinct bodies of work. One body of work stems from small-scale experiments that have directly manipulated species richness (mostly of herbaceous plants in grasslands) and examined how richness influences the annual production of biomass. Hundreds of these studies have been performed, and dozens of meta-analyses have emphasized the consistency among experiments where >80% have shown biomass production is a positive, but decelerating function of species richness.

The second group of studies cited by the authors to establish their claims come from observational studies that have looked at how species richness correlates with environmental gradients that control the productivity of ecosystems, such as gradients in precipitation, temperature, or inorganic nutrients. There continues to be debate over whether species richness is positively correlated with productivity because both the same way to environmental gradients, or whether richness and productivity are related in some non-linear way because they are controlled differently by environmental covariates.

Citing two entirely different bodies of work as evidence that diversity-productivity relationships are highly variable and in need of explanation is a straw-man argument. Imagine I said we need to better understand the highly variable relationship between mass and velocity because cars and feathers have different momentum. The only reason the relationship is highly variable is because cars and feathers have nothing to do with one another. Nor do the two bodies of work cited by the authors to set up the need for their study.

The authors then further their straw-man argument by proceeding with analyses that assume tree species richness is the causal driver of biomass production. They state this in the text (e.g., line 324 "Besides species richness, many other factors can influence forest productivity"), and in their analyses (e.g., Figure S2 and Table S1). Yet, this is the very body of work that has found substantial consistency in how species richness impacts biomass production, and none of this body of work has documented concave-down effects of richness on biomass production.

Bottom line ... I believe the paper inaccurately summarizes the current literature, combining bodies of work that have little to do with one another in order to give the reader the false impression that this field of work has found little consistency. I think this problem could be fixed, but it would take a substantial re-write of the paper.

2. Pattern searching, no mechanisms. When this study is boiled down to its core, it's really little more than an exercise in pattern searching ... looking for statistically significant relationships between tree richness and biomass production in hundreds of general linear models (by environmental covariates, and across ecoregions). There is little attempt to explain why the patterns occur, or why they change across gradients.

The authors do speculate at multiple points in the paper that "The most likely mechanism for the differing BPR types in different climates is that richness and productivity have different relationship with different climate variables ..." (line 83). That's a perfectly reasonable hypothesis, but they never test it. Indeed, I was perplexed as to why the authors did not pursue this, or any other test of a multivariate hypothesis using Structural Equations Modeling, or Hierarchical models to quantify how precipitation and temperature simultaneously influence species richness and biomass production, ultimately leading to a change in the covariance that defines the relationship between the latter two variables. The lack of such analyses is a glaring omission, as it leaves this paper solely in the realm of biological speculation.

Bottom line ... this paper does not offer much insight into, or explanation of, the patterns that are described. As such, it will only serve to confuse, rather than advance, the field. This issue could

potentially be overcome with some more directed analyses of the author's own hypothesis for what is going on in their data, or by replicating the multivariate analyses of people they cite in their paper (e.g., Grace et al. 2016, or see work by Paquette and Messier in *Global Ecology and Biogeography* 2011).

3. Incomplete, inconclusive analyses. Even if one were to accept the pattern-search exercise presented in this paper as being sufficient, the analyses used to quantify the patterns were not sufficient to guarantee robust conclusions. Simply testing whether or not a quadratic term in a polynomial meets some criteria for significance (e.g., $P < 0.05$) does not tell us whether a curvilinear model is an improved fit to the data that adds explanatory power over a linear model. Nor does it tell us whether or not curvilinearity is "internal" to the data (i.e. the 1st derivative falls within the scale of the x-axis) or is being extrapolated beyond the data in-hand. These two issues are very old, and have been discussed at length in the literature on diversity-productivity relationships. It is not ok anymore to simply fit a quadratic model and say whether it is significant. You must show that it is an improved fit over a more parsimonious model, and that it fits the actual data rather than some extrapolation beyond the x-axis.

Bottom line ... this paper does not provide convincing conclusions for the patterns, even with the current analyses. This could be fixed by using model comparison tools (e.g., AIC values and likelihood tests) and performing tests of 'internal humps' for quadratic terms that are significant and that improve fit to data.

Responses to Reviewers:

Reviewer #1 (Remarks to the Author):

Overall the Authors present a very interesting and important study on the biodiversity-productivity relationship in natural forest and the importance of climate as predictor of this relationship. I have a couple of major points that I think could assist with establishing the important context of the study and several other issues that I hope will increase the clarity and depth of ecological meaning that can be drawn by readers.

I am not sure I agree with how the Authors begin framing the novelty of the study. I think there has been a tremendous number of studies investigating tenets of the BPR and a lot of evidence has mounted but the evidence is equivocal especially for more complex ecosystems like forests--- simpler communities such as experimentally derived mesocosms of algae, herbaceous plants etc. have provided I think many generalizable constructs but these have not necessarily held true when then applied and tested in more natural systems etc. I think this is a novel contribution of the Authors study and they should focus on this, and their incorporation of climatic predictors---- bringing a reality check so to speak to this important relationship.

I think the Authors need to re-write their abstract so that it explicitly delineates what the observed divergent relationships have been in natural systems and specifically in forests versus grasslands (where a preponderance of the studies were to date conducted) and then describe explicit what the new findings that this study contributes especially in regards to climatic variation and how precipitation potentially regulates this relationship.

Response 1: Thanks for these constructive suggestions. In the revised version, we incorporated the reviewer's suggestion by emphasizing that there has been a large number of studies

investigating BPRs based on simpler communities such as herbaceous plants, but that the evidence for the nature of these relationships is unclear especially for more complex ecosystems such as forests. We have now also further argued that it is unclear as to what extent BPRs determined for simpler systems can be generalizable or held true when then applied and tested in more natural systems. In doing this we have revised the Abstract (Lines 23-26) and Introduction (Lines 40-52) accordingly.

Line 45 to 47: It would be good to describe briefly the key competing theories...if the Authors study will expand on these ideas we need to see a “thesis” of what they think/interpret these are and I think there should be a focus on natural communities.

Response 2: As suggested, we have expanded the Introduction by focusing more on natural communities (Line 46-52).

Line 54: what are these exceptions? With increasing resources, there is evidence of a selection for more competitive plant species for these resources.

Response 3: In general, we agree that increasing resources favor more competitive species. We have now revised this sentence to make our point clearer and have provided two citations (Line 58-59).

Line 57-58: In natural forest ecosystems but it has been done in grasslands by Grace et al. 2016 and you cite this study earlier...what did they find? Illustrate your points to readers with brief and meaningful examples...your readers should not have to go and read other study to find your examples or what the debates etc are about...the Authors should be illustrating these ideas for readers in a complete study. This is an important study for your paper and the supplementary material includes a very detailed and insightful history of the literature around this topic.

Response 4: We have now made explicit reference to the findings of Grace et al (2016) to increase clarity about this and other relevant concepts and studies (Line 64-69).

Line 59: why the reference to the USA being contiguous...shouldn't the emphasis be on the range of climates etc to fit the purpose of the study.

Response 5: We agree that the emphasis in this sentence should be on the climatic range not on the lower 48 states being contiguous. We have now added “across the large range of temperature and precipitation conditions” to highlight this point (Line 72-73).

Line 59-67: Great dataset...are these all natural forests or were some of these plantations. Logged and then regrown especially given the data is from the US Forest Inventory. When was the data collected---all in the same year or over many years? Are the study sites all different forests? How many species do they hold in common...why was the inventory taken on all of these forests from an ecological stand point what ties them together—is there a management

history in common? I can see now that some of this information is provided in the supplementary material but I think some of it is key for understanding the ecological meaning and robustness of the Author's study and therefore should be presented briefly in the main manuscript.

Response 6: We have now presented a description of several of these requested details in the main text (Line 74 – 80) and in the discussion (Line 190-192). We also added a reference to a detailed description of the range of vegetation included in our study in the Supplementary Fig. 1.

Line 63-67 what about basal area as a measure instead of biomass...can you provide a justification for why you chose this particularly measure of productivity?

Response 7: We chose biomass over basal area for two reasons. First and foremost, as tree height varies dramatically within a site and among sites even for stands with comparable basal area, we maintain that biomass is a better representation of resource acquisition by a forest stand; basal area is two-dimensional and does not incorporate height variability. In addition, most other studies used biomass instead of basal area whenever biomass data are available. The use of biomass instead of basal area here also ensures inter-studies comparison.

Line 73-74: given the complexity of the data and the sampling design, wouldn't it be more robust to use generalized mixed models. This would allow the Authors to repartition the variation in the data according to sites and plots; thereby incorporating the sampling design into the analytical models. I can see that you do use GLMMs later on but why not throughout?

Response 8: We agree that a more robust model would better address BPRs given the complexity of the data. In the revised manuscript, we have now used a hierarchical Bayesian framework (new Fig. 2a) to analyze the BPRs in the climatic space (see Line 247-266 for model details). However, we have also kept the bivariate BPRs to illustrate the range of BPR patterns that have been identified in previous studies. This is because they not only enable a clear comparison of the changes in BPRs with and without the consideration of other factors (new Fig. 1 vs. Fig. 3), but also demonstrate the robustness of the impacts of climate on the BPR patterns.

Line 100-101 Could this relationship found between MAT and productivity when TAP is >1000mm be because there is a little variability in MAT in your dataset at such high precipitation levels. What is the relationship between MAT and TAP?

Response 9: The relationship between MAT and TAP is plotted in Fig. 1. The range for MAT is still quite wide when TAP is > 1000mm, although the variability in MAT is limited when TAP is >1700mm. We have now provided additional analyses of MAT and TAP on richness and productivity (new Fig 4).

Have you thought about using a structural equation model approach like what Grace et al. 2016 Nature used for grasslands?

Response 10: We intended to use a SEM in our initial analysis, but the nature of data and limitation of SEM prevented us to do so for the following two reasons. First, SEM cannot handle non-linear relationships, and our bivariate BPR analysis showed a clear quadratic term of richness on productivity. Second, SEM pools all data into a single model and therefore cannot show the variability of the coefficients across the climate space. Hence, we instead applied a Hierarchical Bayesian framework (HBF) with five sub-models as shown in the new Fig. 2a to include various biotic and abiotic factors that could impact BPRs. Another advantage of the HBF approach is that it can incorporate ecoregions into the model to handle spatial autocorrelation of the data.

Line 143: I don't think you have any way of differentiating between the sampling effect and complementarity because there may be no commonality between species identity amongst the different forest types or are there? Nevertheless, the Authors do not illustrate this for their readers.

Response 11: We completely agree with the reviewer that the challenge of differentiating between the sampling effect and complementarity for across-forest-type or across-climate-range comparative studies lies in the fact that they do not necessarily share a common species pool. We have now explicitly discussed this issue (Line 183-185).

Line 155-169: I think the management implications of this study come from nowhere with providing readers with an understanding of these forests...have they been logged and regrown etc.

Response 12: We have now added several points to the discussion about management implications (Line 186-208). The new results of stand characteristics related influences on richness and productivity provide a better connection to potential management implications.

Line 261 I think the Authors should include a definition for the black color in panel A of Fig. 2 with the legend description provided for all other colors.

Fig. 2: There are quite a number of different predictors in this model. Were predictors scaled? What were the R² values with and without random effects for these models, what were the denominator and numerator degrees of freedom?

Response 13: We have now removed Fig. 2 in the manuscript by incorporating 'ecoregion' as a random factor in the Hierarchical Bayesian Framework (Line 253-258; see Response 10 for details for the justification of HBF), so the points raised in the reviewer's comment no longer apply.

Reviewer #2 (Remarks to the Author):

The main purpose of this study is to understand what drives the form of the Biodiversity-Ecosystem functioning (BEF) relationship which has been one of the main topics in ecology in the past decades. The main hypothesis is that the contrasting results observed in experimental settings and in observational studies may be explained by climatic factors. The results obtained by this study that analyzed natural forests ecosystems in the contiguous United States is that BEF relationship is determined by a combination of total annual precipitation and mean annual temperature, because the relationship between richness and log(productivity) is linear-positive in arid and semi-arid sites, and concave-negative in humid and sub-humid sites. The authors arrived to this conclusion by analyzing data from more than 116,000 plots across forests from which estimations of aerial primary productivity and species richness are available, so this is an observational study and the mechanism behind this pattern cannot be inferred so far. In addition, the ecosystem functioning variable has been log-transformed (log productivity is reported along the manuscript) whereas the richness hasn't, so the inferences relating linear or concave negative functions might be confusing for the broad audience. And the comparison with previous studies need careful at this point.

Response 14: We agree that observational studies have their limitations, but they can provide interesting insights into underlying ecological patterns, which could in turn contribute to greater mechanistic understanding. In addition, it is not possible to conduct a manipulative experiment at this quasi continental scale with such replication. We have now clarified in the main text that productivity measures we used was log-scaled (Line 96). The reason for using log-transformed productivity is that productivity data are not normally distributed. Log transformation helps to bring the data closer to normal distribution to meet statistical assumptions, a practice that has been used in many previous studies including those cited in our manuscript.

The consequences of biodiversity losses on plant production were previously evaluated, but regional studies combining diversity and climate effects are scarce. In addition, the BEF literature was developed based on grasslands experiments involving herbal species, but has rarely been assessed in forests. In synthesis, the study was well designed and the results are compelling and can potentially make an important contribution to the effect of climate on productivity and richness in forests, but the improvement in the knowledge of the BEF relationship is much more limited, and the conclusions cannot be translated to other ecosystems such as grasslands or deserts. I made some suggestions on how to improve the manuscript in my comments below.

Response 15: Thank you for these comments. We have now put more focus on highlighting how our study contributes to the improvement of our understanding of the wider BEF relationship. In particular, our results show that conclusions from highly controlled experiments may have limitations for extrapolation to real-world situations where regional climatic variation may significantly alter observed BEF relationships (Line 156-160). We have now also discussed some limitations of the current study, including that our conclusions may not be immediately translated

to other ecosystem types without further field validation or to other studies with different spatial scales (Line 167-170).

Specific comments

The main subject of this study is the Biodiversity-Ecosystem functioning relationship, which has been mentioned as BEF in the literature (see for example Flynn et al. 2001 or Tilman et al. 2014), even when the main functioning aspect assessed was “primary productivity” and the diversity index was “species richness”. So, I suggest keeping this terminology, changing biodiversity-productivity relationship (BPR) by Biodiversity-Ecosystem functioning (BEF) along the Ms.

Response 16: We decided to keep the use of productivity because usage of the term ‘ecosystem functioning’ also encompasses many other things such as nutrient flux rates, decomposition, and other processes; as such productivity is only one component of ecosystem functioning. However, we also acknowledged the importance of biodiversity in relation with other ecosystem functions in the Discussion of the revised manuscript (Line 194).

#26. I don’t think the following phrase is totally true: "However, we still lack an understanding of what drives the form of the BPR or an underlying theory to explain the varying patterns of BPR across studies". We do have a good understanding and theory of the BEF (see for example Tilman et al 2014 for a synthesis). What we do lack is the influence of climate. And the application of the BEF relationship at larger (regional) scales. I prefer the statement "Nevertheless, to date no large-scale studies have attempted to explicitly address how BPR may change across climatic gradients".

Response 17: Good point. Reviewer 1 and 3 also had similar comment. We have substantially revised the introduction to address this comment (Line 40-52).

#31 "We found that the relationship between richness and log(productivity) is linear..." What does this statement mean in terms of the productivity-richness relationship? How do you translate the log-productivity patter to the untransformed productivity variable?

Response 18: We have now clarified in the text that the relationship is based on log-transformed productivity (Line 96 in the Result section and Line 236-237 in the Methods section). The reason we used log(productivity) is primarily due to the non-normal distribution of productivity, and for this reason, log transformed productivity has also been widely used in most previous BPR studies.

#35 Productivity generally increases with precipitation, while richness has a concave negative relationship with precipitation. It depends on the scale!!! (Chase and Leibold 2002). And on the range of precipitation analyzed (as is acknowledge latter in the Ms). The productivity-precipitation relationships saturated after 1,500 mm/yr precipitation probably because other variables than water limits plant production.

Response 19: We agree that BPRs can be influenced both by grain size and extent as discussed by Chase and Leibold (2002), which was cited in the manuscript. In this study, the grain size (plot) is fixed and extent (10x10% quantile climatic unit) is also relatively fixed. We have now emphasized in our discussion (Line 168-170) that results from this study are relevant only at the spatial scale we considered and that other relationships may emerge at other scales. We also agree that productivity is influenced by many factors other than precipitation. In our revised manuscript, we used a Hierarchical Bayesian Framework (HBF, new Fig. 2a) that includes various biotic and abiotic factors to test BPRs within each of the climatic grid. We have added the new results in Line 107- 152.

References

- Chase, J. M. & Leibold, M. A. Spatial scale dictates the productivity-biodiversity relationship. 179 Nature 416, 427-430 (2002).
- Flynn DFB, Mirotchnick N, Jain M, Palmer MI, Naeem S. 2011. Functional and phylogenetic diversity as predictors of biodiversity-ecosystem-function relationships. Ecology 92:1573–81
- Tilman et al 2014. Biodiversity and Ecosystem Functioning. Annu. Rev. Ecol. Evol. Syst. 45:471–93

We have included these references in our revised manuscript.

Reviewer #3 (Remarks to the Author):

The goal of this study is to determine the cause of variation in diversity-productivity relationship. The authors begin with the claim that diversity-productivity relationships are highly variable across space, sometimes being linear, other times being curvilinear (concave down). They proceed to characterize the diversity-productivity relationships for trees in U.S. forests using estimates of species richness and productivity of trees in 0.01 sq. km plots inventoried for the U.S. Forest Inventory and Analysis (FIA). They then attempt to explain variation in the diversity-productivity relationship based on environmental variables, namely air temperature and precipitation, that are taken from ~1-km gridcells represented in the Global Climate Data database. Lastly, the authors examine how the diversity-productivity relationships vary by across ecoregions of the U.S. (I assume these were EPA Level III ecoregions), presumably because these represent distinct climatic environments.

In my view, the study is presently limited by three problems:

1. Setting up a straw-man argument. To set up their study and justify its importance and contribution to ecology, the authors make two claims: (1) “A wide variety of BPRs have been reported from field studies and meta-analyses” (line 42) and (2) “we still lack an understanding of what drives the form of the BPR or an underlying theory to explain the varying patterns of BPR across studies” (line 26).

As evidence for these two claims, the authors cite papers from two distinct bodies of work. One body of work stems from small-scale experiments that have directly manipulated species

richness (mostly of herbaceous plants in grasslands) and examined how richness influences the annual production of biomass. Hundreds of these studies have been performed, and dozens of meta-analyses have emphasized the consistency among experiments where >80% have shown biomass production is a positive, but decelerating function of species richness.

The second group of studies cited by the authors to establish their claims come from observational studies that have looked at how species richness correlates with environmental gradients that control the productivity of ecosystems, such as gradients in precipitation, temperature, or inorganic nutrients. There continues to be debate over whether species richness is positively correlated with productivity because both the same way to environmental gradients, or whether richness and productivity are related in some non-linear way because they are controlled differently by environmental covariates.

Citing two entirely different bodies of work as evidence that diversity-productivity relationships are highly variable and in need of explanation is a straw-man argument. Imagine I said we need to better understand the highly variable relationship between mass and velocity because cars and feathers have different momentum. The only reason the relationship is highly variable is because cars and feathers have nothing to do with one another. Nor do the two bodies of work cited by the authors to set up the need for their study.

The authors then further their straw-man argument by proceeding with analyses that assume tree species richness is the causal driver of biomass production. They state this in the text (e.g., line 324 “Besides species richness, many other factors can influence forest productivity”), and in their analyses (e.g., Figure S2 and Table S1). Yet, this is the very body of work that has found substantial consistency in how species richness impacts biomass production, and none of this body of work has documented concave-down effects of richness on biomass production.

Bottom line ... I believe the paper inaccurately summarizes the current literature, combining bodies of work that have little to do with one another in order to give the reader the false impression that this field of work has found little consistency. I think this problem could be fixed, but it would take a substantial re-write of the paper.

Response 20: We appreciate the reviewer’s comments. We have addressed these in the following four main aspects.

- 1) We agree that there was scope for improvement in how the body of literature was cited in the old version of our manuscript. To avoid confusion, we now explicitly distinguish the essence of the relevant literature based either on small-scale manipulative experiments or field observations (Line 40-44).
- 2) As suggested by both Reviewers 1 and 2, we clarified the differences between previous research and our current study by emphasizing that previous BPR research has primarily focused on ecosystems with relatively simple structures such as grassland, and that insights regarding more complex natural systems such as forests were limited. We have emphasized this point in the revised version (Line 46-52).
- 3) In addition, as suggested by Reviewer 2, we further highlighted the uniqueness of the study in the sense that our study explicitly addresses how BPRs may change across climatic gradients with the consideration of other biotic and abiotic factors (Line 62-71).

- 4) We have included a new analysis using a Hierarchical Bayesian Framework to study BPRs with consideration of biotic and abiotic factors in addition to MAT and TAP. The results indicated that concave-down BPRs can be observed with or without considering other factors (new Figs 1 & 3), which confirms that the findings of our previous analysis were correct.

2. Pattern searching, no mechanisms. When this study is boiled down to its core, it's really little more than an exercise in pattern searching ... looking for statistically significant relationships between tree richness and biomass production in hundreds of general linear models (by environmental covariates, and across ecoregions). There is little attempt to explain why the patterns occur, or why they change across gradients.

The authors do speculate at multiple points in the paper that "The most likely mechanism for the differing BPR types in different climates is that richness and productivity have different relationship with different climate variables ..." (line 83). That's a perfectly reasonable hypothesis, but they never test it. Indeed, I was perplexed as to why the authors did not pursue this, or any other test of a multivariate hypothesis using Structural Equations Modeling, or Hierarchical models to quantify how precipitation and temperature simultaneously influence species richness and biomass production, ultimately leading to a change in the covariance that defines the relationship between the latter two variables. The lack of such analyses is a glaring omission, as it leaves this paper solely in the realm of biological speculation.

Bottom line ... this paper does not offer much insight into, or explanation of, the patterns that are described. As such, it will only serve to confuse, rather than advance, the field. This issue could potentially be overcome with some more directed analyses of the author's own hypothesis for what is going on in their data, or by replicating the multivariate analyses of people they cite in their paper (e.g., Grace et al. 2016, or see work by Paquette and Messier in *Global Ecology and Biogeography* 2011).

Response 21: We appreciate the constructive criticism. In the revised manuscript, we conducted an extensive new analysis using a Hierarchical Bayesian Framework (HBF) with five sub-models (see new Fig. 2a for model structure) to test whether richness and productivity have different relationship with different climate variables, we also included other biotic and abiotic factors to make the analysis more comprehensive.

Regarding the reviewer's suggestion to use Structural Equations Modeling (SEM), we had indeed intended to use a SEM in our initial analysis. But the nature of the data and limitations of SEM prevented us from doing so for the following two reasons. First, SEM cannot handle non-linear relationships, and our bivariate BPR analysis showed a clear quadratic term of richness on productivity. Second, SEM pools all data into a single model and therefore cannot show the variability of the coefficients across the climate space. Hence, we instead applied a HBF to include various biotic and abiotic factors that could impact BPRs. Another advantage of the HBF approach is that it can incorporate ecoregions into the model to handle spatial autocorrelation of the data.

Our new HBF based analysis has indeed greatly improved our understanding of how climate impacts BPRs (new Figs. 4 and 5). We have now also added substantial new text explaining the results of the new analysis and the possible reasons behind these varying BPRs across the climatic space. The HBF results indicated that climate directly affected the observed BPR patterns through differing associations of climatic variables with richness versus with productivity in different regions of the climatic space (Line 124-137). It also suggested that climate impacts BPRs indirectly through site characteristics (Line 138-152).

3. Incomplete, inconclusive analyses. Even if one were to accept the pattern-search exercise presented in this paper as being sufficient, the analyses used to quantify the patterns were not sufficient to guarantee robust conclusions. Simply testing whether or not a quadratic term in a polynomial meets some criteria for significance (e.g., $P < 0.05$) does not tell us whether a curvilinear model is an improved fit the data that adds explanatory power over a linear model. Nor does it tell us whether or not curvilinearity is “internal” to the data (i.e. the 1st derivative falls within the scale of the x-axis) or is being extrapolated beyond the data in-hand. These two issues are very old, and have been discussed at length in the literature on diversity-productivity relationships. It is not ok anymore to simply fit a quadratic model and say whether it is significant. You much show that it is an improved fit over a more parsimonious model, and that it fits the actual data rather than some extrapolation beyond the x-axis.

Bottom line ... this paper does not provide convincing conclusions for the patterns, even with the current analyses. This could be fixed by using model comparison tools (e.g., AIC values and likelihood tests) and performing tests of ‘internal humps’ for quadratic terms that are significant and that improve fit to data.

Response 22: Thanks for the suggestions. In the revised version, we have now analyzed BPRs in two different ways. We first analyzed the bivariate BPRs. Instead of using p-value (as we used in the old version) for the quadratic term to determine the relationship pattern, we have now used AIC values as suggested by the reviewer. We have distinguished if the relationship with a significant quadratic term is hump-shape within the data range (by checking if the 1st derivative falls within the scale of the x-axis; see new Fig. 1 and Supplementary Fig. 2 for the updated results). Moreover, we have now further investigated the BPRs using a HBF (new Fig. 2). The new HBF-based results have further confirmed the impacts of climate on the observed BPR patterns (new Fig. 3). These analyses have been explained in the revised Methods (Line 247-266) and Results (Line 108-123) sections.

Reviewers' comments:

Reviewer #1 (Remarks to the Author):

Thank you to the Authors for their careful responses and actions to address the suggestions made in the previous round of reviews. The Authors have impressively re-analysed the data using a much more robust methodology and have re-written large chunks of the text. I think this has increased the significance of the study and its clarity and it is my opinion that this is an important contribution to the BEF debate. I hope this study receives the attention that it deserves.

Reviewer #2 (Remarks to the Author):

The main purpose of this study is to understand what drives the form of the Biodiversity-Ecosystem functioning (BEF) relationship which has been one of the main topics in ecology in the past decades. The present version of the manuscript has improved substantially in comparison with the original version, and the authors had incorporated the big majority of the reviewers' comments. For example, while the original version of the manuscript did not address appropriately other factors apart from climate, in this version of the manuscript, the authors analyzed the data with a hierarchical Bayesian modeling approach that include biotic variables (such as stand age, stem density, succession stage...). The results changed accordingly. However, the message in the title and the abstract remained pretty much the same. For instance, the new Bayesian approach revealed that other site characteristics such as stand age changed the BPRs function in dry climates, so it is not only mean climate that determines the biodiversity-productivity relationship in natural forests in the broad sense. This should be carefully revised across the text. Finally, there are some minor comments below that I suggest you consider.

Specific comments to the authors

-As the main results from the manuscript have changed, so do the message in the title and the abstract. For instance, the new Bayesian hierarchical approach revealed that "when considering other site characteristics such as stand age, BPRs changed to neutral in dry climates.....", so it is not only climate that determines the biodiversity-productivity relationship in natural forests. This should be carefully revised across the text.

-I don't see the point that forest ecosystems are more complex than grasslands... I do agree that most BEF theory was developed in artificial assemblies of aquatic and herbaceous-dominated ecosystems, probably because non-woody vegetation is easier to manipulate. However, the BEF in forest ecosystems is expected to be the same as in grasslands. What is expected to differ are "simple or artificial assemblies" from "natural ecosystems", independent of biome types (grasslands or forests).

-The main result of this study which states that the bivariate BPRs were positive in dry climates and hump-shaped in mesic climates is not related with "climate variability". So please change the sentence "Our findings indicate that climatic variation is an underlying determinant for contrasting BPRs observed among previous studies" (line 31) by one that refers to climate type (arid vs. mesic).

-The last sentence of the abstract that state "These findings suggest that different management strategies are needed to promote biodiversity and productivity in different climatic conditions." (line 34) is not a conclusion of the results described above. Which are the management strategies recalled here? Are you referring to stem density? This should be clarified.

Reviewer #4 (Remarks to the Author):

As per the directions of the editor, I am reviewing the Bayesian hierarchical model component of this paper. The authors have developed a hierarchical model for:

1. productivity, as a function of MAT, TAP, stem density, stand age, soil C:N, richness, richness²
2. richness, as a function of MAT, TAP, stem density, stand age, soil C:N
3. stem density, stand age, soil C:N as a function of MAT and TAP

Assuming the hierarchical structure reflects current understanding of relevant ecological processes, a Bayesian hierarchical model would be well suited to modelling productivity and richness. However, there are some considerations.

Firstly, it is stated on line 254:255 that “we added ecoregions in the submodels as a random intercept to account for spatial autocorrelation”. However, the terms are normal with zero mean and what appears to be shared variance among plots belonging to the same ecoregion. Is this correct? If so, there are a few issues and/or clarifications needed for this statement:

1. The term $N(0, \sigma^2_{\text{ecoregion}})$ implies an error term rather than a random intercept as a Gaussian random variable with zero mean typically models noise. A random intercept is typically drawn from a normal with *non-zero* mean – this helps with model identifiability.
2. Specifying a common variance value across multiple plots does *not* account for spatial autocorrelation. What you are interested in is how two plots within the same ecoregion vary with each other rather than how one plot varies by itself. Typically, spatial correlation is accounted for with covariance structures defined for some *neighbourhood* of plots, such as within an ecoregion. Often, this spatial aspect is captured with *multi-variate* normal distributions such as the CAR model. The text and notation seems to suggest that a spatial covariance structure was not implemented in this model.

A common way to model spatial autocorrelation is with the Conditional AutoRegressive (CAR) model. You can implement this quite easily with GeoBUGS, which comes with WinBUGS, or manually in JAGS (see for example www2.stat.duke.edu/~cr173/Sta444_Sp17/slides/Lec19.pdf).

Secondly, although model fit results were reported for the GLM, it does not appear to be reported for the hierarchical models. This is critical in establishing confidence in the results of the model. Measures such as deviance and the mean squared error in richness and productivity would be helpful to understand how well the models are fitting.

Additionally, when reading the Methods and Supplementary materials, it is not clear how the type of relationship between productivity and richness is inferred using the hierarchical model. Subsequent reading reveals that it is explained in the caption for Fig 3, but it should be made clear in the methods that it is based on the posterior distributions for β_{17} and β_{18} . Please clarify.

Finally, there is a potential concern about model identifiability given the number of terms and more importantly their replication in the hierarchy and the possibility of introducing collinearity. Model identifiability could impact your estimates of β_{17} and β_{18} and hence the findings. Upon re-reading, it appears that variables such as stem density, stand age, soil C:N, richness and productivity were observed? Is this the case? Or was there a lot of missing data? If the data is fairly dense for these variables, the model could be simplified (i.e. reduce the number of β terms to estimate) by using for example stem density, stand age and soil C:N as predictor variables directly in the model.

Note: I am assuming you for stem density, stand age and soil C:N that these variables are non-negative and that you have used truncated normals (e.g. $\text{dnorm}(0, 0.001); T(0, 1)$ using rjags)

Response to the Decision on manuscript NCOMMS-18-03032A

Note that our responses to the comments are in blue font.

Reviewer #1 (Remarks to the Author):

Thank you to the Authors for their careful responses and actions to address the suggestions made in the previous round of reviews. The Authors have impressively re-analysed the data using a much more robust methodology and have re-written large chunks of the text. I think this has increased the significance of the study and its clarity and it is my opinion that this is an important contribution to the BEF debate. I hope this study receives the attention that it deserves.

Response 1: Thanks for the nice comments. We appreciate your remark that our work adds an important contribution to the BEF debate.

Reviewer #2 (Remarks to the Author):

The main purpose of this study is to understand what drives the form of the Biodiversity-Ecosystem functioning (BEF) relationship which has been one of the main topics in ecology in the past decades. The present version of the manuscript has improved substantially in comparison with the original version, and the authors had incorporated the big majority of the reviewers' comments. For example, while the original version of the manuscript did not address appropriately other factors apart from climate, in this version of the manuscript, the authors analyzed the data with a hierarchical Bayesian modeling approach that include biotic variables (such as stand age, stem density, succession stage...). The results changed accordingly. However, the message in the title and the abstract remained pretty much the same. For instance, the new Bayesian approach revealed that other site characteristics such as stand age changed the BPRs function in dry climates, so it is not only mean climate that determines the biodiversity-productivity relationship in natural forests in the broad sense. This should be carefully revised across the text. Finally, there are some minor comments below that I suggest you consider.

Response 2: We agree that the new hierarchical Bayesian models have provided new insights into the BEF relationships. The reason why we have kept a similar title and most of the abstract is to emphasize the importance of considering climatic variables in large-scale BEF analyses. Climatic variables play important roles at two levels: 1) they help the partitioning of the linear vs hump-shaped BEF relationships across climatic space, and 2) they help to explain various relationships between productivity and other site characteristics within each climatic unit. However, we agree with the reviewer that there was scope for providing a more comprehensive picture of the results. We have now clarified in the abstract (Lines 30-34) and in the discussion (Lines 167-171) the importance of climatic variables and other biotic factors (e.g., stand age and density) and abiotic factors (e.g., soil properties) in influencing BEF relationships.

Specific comments to the authors

-As the main results from the manuscript have changed, so do the message in the title and the abstract. For instance, the new Bayesian hierarchical approach revealed that “when considering other site characteristics such as stand age, BPRs changed to neutral in dry climates....”, so it is not only climate that determines the biodiversity-productivity relationship in natural forests. This should be carefully

revised across the text.

Response 3: Revised as suggested (see Response 2 for details)

-I don't see the point that forest ecosystems are more complex than grasslands... I do agree that most BEF theory was developed in artificial assemblies of aquatic and herbaceous-dominated ecosystems, probably because non-woody vegetation is easier to manipulate. However, the BEF in forest ecosystems is expected to be the same as in grasslands. What is expected to differ are "simple or artificial assemblies" from "natural ecosystems", independent of biome types (grasslands or forests).

Response 4: We agree that both grasslands and forests are complex systems, although forest ecosystems are more complex in terms of their structure than grasslands as they often contain an herbaceous/shrub layer, an understory, a mid-story, and an overstory. Regardless, system complexity is not a focus of this manuscript. Therefore, we have now removed the complexity argument (Lines 46-47).

-The main result of this study which states that the bivariate BPRs were positive in dry climates and hump-shaped in mesic climates is not related with "climate variability". So please change the sentence "Our findings indicate that climatic variation is an underlying determinant for contrasting BPRs observed among previous studies" (line 31) by one that refers to climate type (arid vs. mesic).

Response 5: Changed as suggested (Lines 30-31)

-The last sentence of the abstract that state "These findings suggest that different management strategies are needed to promote biodiversity and productivity in different climatic conditions." (line 34) is not a conclusion of the results described above. Which are the management strategies recalled here? Are you referring to stem density? This should be clarified.

Response 6: We have now revised the concluding remark to read: "These findings suggest that tradeoffs need be taken into account when considering whether to maximize productivity vs. conserve biodiversity, especially in mesic climates." (Lines 32-34)

Reviewer #4 (Remarks to the Author):

As per the directions of the editor, I am reviewing the Bayesian hierarchical model component of this paper. The authors have developed a hierarchical model for:

1. productivity, as a function of MAT, TAP, stem density, stand age, soil C:N, richness, richness²
2. richness, as a function of MAT, TAP, stem density, stand age, soil C:N
3. stem density, stand age, soil C:N as a function of MAT and TAP

Assuming the hierarchical structure reflects current understanding of relevant ecological processes, a Bayesian hierarchical model would be well suited to modelling productivity and richness. However, there are some considerations.

Firstly, it is stated on line 254:255 that "we added ecoregions in the submodels as a random intercept to account for spatial autocorrelation". However, the terms are normal with zero mean and what appears to be shared variance among plots belonging to the same ecoregion. Is this correct? If so, there are a few issues and/or clarifications needed for this statement:

1. The term $N(0, \sigma^2_{\text{ecoregion}})$ implies an error term rather than a random intercept as a

Gaussian random variable with zero mean typically models noise. A random intercept is typically drawn from a normal with *non-zero* mean – this helps with model identifiability.

Response 7: Thank you for your suggestion. We have now revised the code to draw a random intercept from a normal distribution with a ‘non-zero’ mean as suggested. The results from the revised analyses are very similar to our earlier analyses as indicated in Lines 134-137 and copied below (red numbers indicate model results from previous analyses (crossed out) and the revised analyses (not crossed out). These minor changes leave our conclusions entirely unchanged.

“Overall, there were more climatic units that had statistically significant associations (at $P < 0.05$) between MAT and richness (~~30~~ 31 units) and between MAT and productivity (~~36~~ 35 units) than between TAP and richness (22 units) and between TAP and productivity (~~24~~ 23 units)”.

2. Specifying a common variance value across multiple plots does *not* account for spatial autocorrelation. What you are interested in is how two plots within the same ecoregion vary with each other rather than how one plot varies by itself. Typically, spatial correlation is accounted for with covariance structures defined for some *neighbourhood* of plots, such as within an ecoregion. Often, this spatial aspect is captured with *multi-variate* normal distributions such as the CAR model. The text and notation seems to suggest that a spatial covariance structure was not implemented in this model. A common way to model spatial autocorrelation is with the Conditional AutoRegressive (CAR) model. You can implement this quite easily with GeoBUGS, which comes with WinBUGS, or manually in JAGS (see for example www2.stat.duke.edu/~cr173/Sta444_Sp17/slides/Lec19.pdf).

Response 8: Thanks for pointing out the random intercept issue, which implies that there is common variance value across multiple plots within the same ecoregion. Our statement that “we added ecoregions in the sub-models as a random intercept to account for spatial autocorrelation” in the previous version is imprecise. The reason we used a random intercept (common variance within an ecoregion) in our analysis is that we aggregated our data in the climatic space (not the geographic space); therefore, plots from different ecoregions could be pooled into the same climatic unit. In other words, there could be heterogeneities within each climatic space because plots from different ecoregions are likely associated with different sets of latent variables (e.g., different land use or disturbance histories). As argued in our earlier paper (Dixon Hamil et al. 2016), analyses without considering this type of heterogeneity could lead to incorrect inferences. Therefore, we used the random intercept as a way to account for heterogeneity among plots that could be inherently linked to different ecoregions. To avoid possible confusions, we have now changed the statement to read “we added ecoregions in the sub-models as a random intercept to account for the spatial heterogeneity between geographically distant plots that share a similar climate” (Lines 265-267).

Dixon Hamil, K.A., Iannone III, B.V., Huang, W.K. et al. *Landscape Ecol* (2016) 31: 7.
<https://doi.org/10.1007/s10980-015-0288-z>

Secondly, although model fit results were reported for the GLM, it does not appear to be reported for the hierarchical models. This is critical in establishing confidence in the results of the model. Measures such as deviance and the mean squared error in richness and productivity would be helpful to understand how well the models are fitting.

Response 9: We agree regarding the importance of providing model fit results. We have now added deviance information for each model in our supplementary materials (Supplementary Table 2).

Additionally, when reading the Methods and Supplementary materials, it is not clear how the type of relationship between productivity and richness is inferred using the hierarchical model. Subsequent reading reveals that it is explained in the caption for Fig 3, but it should be made clear in the methods that it is based on the posterior distributions for β_{17} and β_{18} . Please clarify.

Response 10: Thanks. We have now added the details in the Methods section (Lines 256-262 and Lines 275-276).

Finally, there is a potential concern about model identifiability given the number of terms and more importantly their replication in the hierarchy and the possibility of introducing collinearity. Model identifiability could impact your estimates of β_{17} and β_{18} and hence the findings. Upon re-reading, it appears that variables such as stem density, stand age, soil C:N, richness and productivity were observed? Is this the case? Or was there a lot of missing data? If the data is fairly dense for these variables, the model could be simplified (i.e. reduce the number of β terms to estimate) by using for example stem density, stand age and soil C:N as predictor variables directly in the model. Note: I am assuming you for stem density, stand age and soil C:N that these variables are non-negative and that you have used truncated normals (e.g. $\text{dnorm}(0, 0.001); T(0,1)$ using rjags)

Response 11: We appreciate the reviewer's comments about model identifiability. We constructed another model using stem density, stand age and soil C:N as direct predictor variables (i.e., without β_{1-6}). The model inferences are very similar regardless of whether or not we include these terms as direct predictors (see table below without β_{1-6} and Supplementary Table 2 with β_{1-6} for comparisons). We retained the full model in the manuscript as most ecology readers would be interested to see the full associations among the variables. All the variables mentioned by the reviewer are non-negative and we used standardized values $(x_i - \bar{x})/\sigma$ of these variables in our analyses.

Climatic space defined in Fig. 1			Sub-models in the full hierarchical model described in Fig. 2a*												
MAT range (°C)	TAP range (mm)	Plot n	Richness ~ $Pois(\mu_{Richness_i})$					Productivity ~ $N(\mu_{Productivity_i}, \sigma^2_{Productivity})$							
			β_7	β_8	β_9	β_{10}	β_{11}	β_{12}	β_{13}	β_{14}	β_{15}	β_{16}	β_{17}	β_{18}	
(3.4, 4.5)	(721, 827)	2781	0.067 (0.023, 0.11)	0.037 (-0.023, 0.099)	0.488 (0.444, 0.532)	0.101 (0.059, 0.144)	-0.036 (0.08, 0.008)	0.039 (0.045, 0.122)	-0.028 (0.148, 0.092)	1.267 (1.185, 1.349)	-0.295 (0.371, 0.219)	0.059 (0.029, 0.147)	0.581 (0.494, 0.668)	-0.552 (-0.494, 0.442)	
(3.4, 4.5)	(829, 1015)	1456	0.056 (0.003, 0.114)	-0.007 (0.021, 0.273)	0.392 (0.336, 0.448)	0.078 (0.024, 0.133)	-0.002 (0.065, 0.061)	0.041 (0.064, 0.146)	0.023 (0.188, 0.236)	0.956 (0.859, 1.053)	0.053 (0.037, 0.144)	-0.144 (0.259, 0.029)	0.317 (0.215, 0.419)	-0.394 (-0.534, 0.256)	
(3.4, 4.5)	(1017, 1124)	489	0.056 (0.034, 0.147)	-0.007 (0.086, 0.073)	0.319 (0.233, 0.406)	0.061 (0.028, 0.15)	0.009 (0.091, 0.11)	0.203 (0.05, 0.356)	0.188 (0.054, 0.322)	0.711 (0.567, 0.856)	0.14 (-0.007, 0.288)	0.114 (0.021, 0.25)	0.156 (0.015, 0.326)	-0.156 (-0.264, 0.189)	
(3.4, 4.5)	(1126, 1208)	198	0 (-0.137, 0.136)	-0.118 (-0.253, 0.016)	0.454 (0.301, 0.612)	-0.085 (-0.254, 0.088)	0.012 (0.174, 0.226)	-0.011 (0.242, 0.219)	0.19 (-0.029, 0.409)	1.002 (0.745, 1.26)	0.014 (0.249, 0.278)	-0.09 (0.367, 0.186)	0.117 (0.201, 0.435)	-0.421 (-0.728, 0.112)	
(3.4, 4.5)	(1210, 1327)	252	0.067 (0.053, 0.187)	-0.043 (0.163, 0.077)	0.074 (0.058, 0.206)	-0.053 (0.184, 0.08)	0.068 (0.1, 0.262)	0.109 (0.045, 0.264)	-0.059 (0.215, 0.095)	0.308 (0.143, 0.473)	0.057 (0.11, 0.222)	0.079 (0.12, 0.276)	0.183 (0.001, 0.364)	-0.133 (-0.362, 0.097)	
(3.4, 4.5)	(1329, 1465)	184	0.154 (0.011, 0.298)	-0.039 (0.199, 0.123)	0.04 (-0.108, 0.189)	-0.09 (-0.274, 0.099)	-0.227 (-0.471, 0.072)	0.127 (0.1, 0.353)	-0.049 (0.283, 0.185)	0.122 (0.102, 0.344)	0.306 (0.04, 0.57)	0.164 (0.205, 0.5)	0.16 (-0.103, 0.424)	0.104 (-0.219, 0.426)	
(3.4, 4.5)	(1471, 2971)	395	0.044 (-0.063, 0.151)	-0.006 (-0.129, 0.116)	0.257 (0.138, 0.376)	0.003 (0.111, 0.116)	0.056 (0.067, 0.181)	0.028 (0.169, 0.225)	-0.227 (-0.45, 0.002)	1.159 (0.944, 1.373)	0.356 (0.151, 0.56)	-0.099 (-0.321, 0.122)	0.324 (0.088, 0.56)	-0.421 (-0.711, 0.132)	
(-0.7, 3.2)	(279, 420)	569	-0.047 (0.182, 0.082)	-0.105 (0.026, 0.235)	0.208 (0.077, 0.342)	-0.022 (0.149, 0.106)	0.085 (0.036, 0.206)	-0.049 (-0.237, 0.136)	0.061 (0.112, 0.236)	1.246 (1.074, 1.42)	-0.702 (0.871, 0.534)	0.289 (0.1, 0.47)	0.162 (0.018, 0.341)	0.037 (-0.198, 0.273)	
(-2.9, 3.2)	(422, 569)	3109	0.048 (0.008, 0.103)	0.018 (0.03, 0.066)	0.332 (0.278, 0.386)	0.101 (0.047, 0.156)	0.037 (0.017, 0.092)	0.253 (0.17, 0.337)	0.092 (0.022, 0.162)	1.164 (1.088, 1.24)	-0.468 (0.541, 0.395)	0.019 (0.061, 0.1)	0.31 (0.234, 0.385)	-0.16 (-0.257, 0.063)	
(-3.2, 3.2)	(571, 719)	4458	0.089 (0.028, 0.148)	0.055 (0.014, 0.095)	0.447 (0.405, 0.489)	0.044 (0.002, 0.086)	-0.02 (-0.064, 0.024)	0.412 (0.319, 0.506)	-0.056 (-0.124, 0.011)	1.509 (1.439, 1.58)	-0.703 (0.771, 0.635)	-0.179 (-0.255, 0.102)	0.886 (0.803, 0.968)	-0.331 (-0.411, 0.251)	
(-3.1, 3.2)	(721, 827)	1478	0.19 (0.079, 0.3)	0.008 (0.06, 0.077)	0.374 (0.305, 0.442)	0.038 (0.031, 0.108)	0.099 (0.021, 0.178)	0.779 (0.591, 0.969)	0.104 (0.007, 0.215)	1.313 (1.201, 1.425)	-0.306 (0.417, 0.197)	0.14 (0.005, 0.274)	0.582 (0.452, 0.712)	-0.25 (-0.392, 0.107)	
(-2.4, 3.2)	(829, 1015)	882	0.234 (0.075, 0.391)	-0.003 (0.14, 0.131)	0.354 (0.277, 0.431)	0.082 (0.001, 0.162)	0.031 (0.114, 0.178)	0.281 (0.05, 0.515)	0.165 (0.042, 0.374)	1.044 (0.903, 1.185)	0.211 (0.076, 0.347)	0.022 (0.198, 0.241)	0.728 (0.526, 0.928)	-0.37 (-0.604, 0.136)	
(-1.6, 3.2)	(1017, 1123)	241	0.114 (0.049, 0.277)	-0.118 (0.239, 0.004)	0.418 (0.27, 0.568)	0.128 (0.03, 0.287)	-0.004 (0.182, 0.164)	0.416 (0.119, 0.717)	-0.076 (0.311, 0.158)	0.832 (0.549, 1.116)	0.443 (0.162, 0.724)	-0.024 (0.324, 0.02)	0.577 (0.369, 0.869)	-0.558 (-0.901, 0.218)	
(-1.5, 3.2)	(1126, 1207)	109	0.203 (0.038, 0.446)	0.067 (0.117, 0.252)	0.253 (0.06, 0.452)	-0.105 (0.314, 0.11)	0.058 (0.198, 0.3)	0.216 (0.236, 0.672)	-0.338 (0.708, 0.032)	0.94 (0.547, 1.331)	0.02 (-0.403, 0.435)	0.094 (0.385, 0.557)	0.553 (0.078, 1.03)	-0.288 (-0.926, 0.354)	
(-0.1, 3.2)	(1212, 1322)	84	0.178 (0.072, 0.433)	0.007 (0.22, 0.235)	0.217 (0.348, 0.463)	-0.059 (0.348, 0.251)	0.256 (0.256, 0.67)	0.579 (0.197, 0.963)	0.138 (0.212, 0.488)	0.662 (0.267, 1.061)	0.176 (0.245, 0.596)	-0.272 (0.807, 0.262)	0.087 (0.333, 0.512)	-0.222 (-0.884, 0.428)	
(0.4, 3.2)	(1329, 1466)	102	0.261 (0.011, 0.538)	-0.082 (0.307, 0.141)	-0.05 (-0.339, 0.242)	0.03 (-0.292, 0.374)	-0.099 (0.406, 0.216)	0.254 (0.109, 0.616)	-0.272 (0.584, 0.04)	1.221 (0.791, 1.645)	0.553 (0.134, 0.967)	0.19 (0.176, 0.549)	0.87 (0.423, 1.311)	-0.315 (-0.939, 0.311)	
(-0.8, 3.2)	(1470, 2931)	314	0.171 (0.022, 0.319)	0.065 (0.097, 0.235)	0.351 (0.193, 0.51)	-0.074 (0.21, 0.062)	0.05 (0.127, 0.248)	0.601 (0.323, 0.88)	-0.13 (0.434, 0.176)	2.075 (1.77, 2.377)	0.397 (0.134, 0.66)	0.037 (0.328, 0.402)	0.224 (0.086, 0.533)	-0.098 (-0.421, 0.226)	

* N , normal distribution; $Pois$, Poisson distribution; μ , mean; σ^2 , variance; α , intercept; β , slope; non-informative priors were used for intercepts (α s), slope coefficients (β s), and random intercept ($\mu_{ecoregion}$) from a normal distribution of mean = 0 and variance = 1,000

** An estimate of expected predictive error (lower deviance is better) in JAGS (a program for analysis of Bayesian hierarchical models using Markov Chain Monte Carlo simulation)⁴.

REVIEWERS' COMMENTS:

Reviewer #4 (Remarks to the Author):

Thank you for addressing my comments. The refinements made to the Bayesian hierarchical model and clarification of the details of its implementation (e.g. spatial heterogeneity) really help to build a solid statistical analysis of the data. Well done! Thank you for being transparent and for the very helpful highlighting of changes in the response and the paper.

REVIEWERS' COMMENTS:

Reviewer #4 (Remarks to the Author):

Thank you for addressing my comments. The refinements made to the Bayesian hierarchical model and clarification of the details of its implementation (e.g. spatial heterogeneity) really help to build a solid statistical analysis of the data. Well done! Thank you for being transparent and for the very helpful highlighting of changes in the response and the paper.

Response: Thanks for the review and generous comments